# The Effects of Probiotics on Small Intestinal Microbiota Composition, Inflammatory Cytokines and Intestinal Permeability in Patients with Non-Alcoholic Fatty Liver Disease

**DOI:** 10.3390/biomedicines11020640

**Published:** 2023-02-20

**Authors:** Nurainina Ayob, Khairul Najmi Muhammad Nawawi, Mohamad Hizami Mohamad Nor, Raja Affendi Raja Ali, Hajar Fauzan Ahmad, Seok Fang Oon, Norfilza Mohd Mokhtar

**Affiliations:** 1Department of Physiology, Faculty of Medicine, Universiti Kebangsaan Malaysia, Kuala Lumpur 56000, Malaysia; 2Gastroenterology and Hepatology Unit, Department of Medicine, Faculty of Medicine, Universiti Kebangsaan Malaysia, Kuala Lumpur 56000, Malaysia; 3GUT Research Group, Faculty of Medicine, Universiti Kebangsaan Malaysia, Kuala Lumpur 56000, Malaysia; 4School of Medical and Life Sciences, Sunway University, Petaling Jaya 47500, Malaysia; 5Faculty of Industrial Sciences and Technology, Lebuhraya Tun Razak, Kuantan 26300, Malaysia; 6B-Crobes Laboratory Sdn. Bhd., Ipoh 31400, Malaysia

**Keywords:** NAFLD, probiotics, gut microbiota, inflammatory cytokines, intestinal permeability, tight junction, GLA

## Abstract

The prevalence of non-alcoholic fatty liver disease (NAFLD) has soared globally. As our understanding of the disease grows, the role of the gut-liver axis (GLA) in NAFLD pathophysiology becomes more apparent. Hence, we focused mainly on the small intestinal area to explore the role of GLA. We looked at how multi-strain probiotics (MCP^®^ BCMC^®^ strains) containing six different *Lactobacillus* and *Bifidobacterium* species affected the small intestinal gut microbiota, inflammatory cytokines, and permeability in NAFLD patients. After six months of supplementation, biochemical blood analysis did not show any discernible alterations in either group. Five predominant phyla known as Actinobacteria, Proteobacteria, Firmicutes, Bacteroidota and Fusobacteria were found in NAFLD patients. The probiotics group demonstrated a significant cluster formation of microbiota composition through beta-diversity analysis (*p* < 0.05). This group significantly reduced three unclassifiable species: *unclassified_Proteobacteria, unclassified_Streptococcus,* and *unclassified_Stenotrophomonas*. In contrast, the placebo group showed a significant increase in *Prevotella_melaninogenica* and *Rothia_mucilaginosa*, which were classified as pathogens. Real-time quantitative PCR analysis of small intestinal mucosal inflammatory cytokines revealed a significant decrease in IFN-γ (−7.9 ± 0.44, *p <* 0.0001) and TNF-α (−0.96 ± 0.25, *p <* 0.0033) in the probiotics group but an increase in IL-6 (12.79 ± 2.24, *p <* 0.0001). In terms of small intestinal permeability analysis, the probiotics group, unfortunately, did not show any positive changes through ELISA analysis. Both probiotics and placebo groups exhibited a significant increase in the level of circulating zonulin (probiotics: 107.6 ng/mL ± 124.7, *p* = 0.005 vs. placebo: 106.9 ng/mL ± 101.3, *p* = 0.0002) and a significant decrease in circulating zonula occluden-1 (ZO-1) (probiotics: −34.51 ng/mL ± 18.38, *p* < 0.0001 vs. placebo: −33.34 ng/mL ± 16.62, *p* = 0.0001). The consumption of *Lactobacillus* and *Bifidobacterium* suggested the presence of a well-balanced gut microbiota composition. Probiotic supplementation improves dysbiosis in NAFLD patients. This eventually stabilised the expression of inflammatory cytokines and mucosal immune function. To summarise, more research on probiotic supplementation as a supplement to a healthy diet and lifestyle is required to address NAFLD and its underlying causes.

## 1. Introduction

Non-alcoholic fatty liver disease (NAFLD) is on the rise, as are obesity and metabolic diseases such as hypertension, type-2-diabetes mellitus (T2DM) and dyslipidemia [1]. NAFLD is a spectrum disorder that begins with the accumulation of fat in liver cells known as steatosis and progresses to steatohepatitis, which occurs when the liver is inflamed [2]. Cirrhosis is the final stage of this disease in which healthy liver tissue is completely replaced by scarred tissue, resulting in permanent liver damage. Generally, an unbalanced diet combined with a sedentary lifestyle is the primary cause of NAFLD [3]. The gut microbiota plays various important roles in the pathogenesis of NAFLD. The symbiotic relationship between microbiota within the intestine is maintained by several essential functions such as vitamin synthesis, resistance from pathogens to colonize the intestine, digestion and also maintenance of the gut-liver axis (GLA) maintenance [4]. GLA describes the close anatomical and functional interaction between the liver and gastrointestinal tract which affects the gut microbiome and body immune system [5,6]. Two key elements of this complex axis are the intestinal barrier and gut microbiota and changes in either of them may accelerate the onset of liver damage [7]. The changes include small intestinal bacteria overgrowth (SIBO), dysbiosis and increased intestinal permeability which is also referred to as leaky gut [8].

Dysbiosis refers to the imbalance between the population of normal microbiota and pathogenic microbiota. Dysbiosis causes the secretion of toxins into the liver through the portal vein due to factors that increase the permeability of the intestinal border. In this context, the relationship of the GLA structure in the formation of NAFLD is important. The GLA that covers the border of the intestinal lining is responsible for controlling the translocation of products produced by the intestinal microbiota [9]. Disturbances in homeostasis cause intestinal barrier breakage which eventually fosters “bacterial translocation” [10]. As a result, the presence of dysbiosis at an early stage which causes an increased translocation of toxins and inflammatory factors will alter the immunological effects in the patient’s body, leading to the secretion of pro-inflammatory cytokines [11]. The emphasis on the small intestinal area is critical in understanding the role of GLA in the pathogenesis of NAFLD. The relationship between the liver and microbiota exists through the presence of the intestinal mucosa layer including intestinal epithelial cells (IECs) that function to maintain intestinal homeostasis.

Currently, no drugs have been approved by the Food and Drug Administration (FDA) to treat NAFLD [12,13,14]. A balanced diet, weight loss, and consistent physical activity, on the other hand, is a strategy for controlling the progression of this disease [15]. Many pharmacological studies have been developed to pave the way for new therapeutic discoveries. Alternative treatments include the use of drugs such as anti-diabetics and anti-oxidants, which have been shown to reduce fat accumulation in liver cells and inhibit the induction of oxygen radical species [16,17,18]. In addition to the rapid development of pharmacological studies, researchers began to focus on the manipulation of the gut microbiota as an alternative treatment almost two decades ago [19]. This manipulation is possible due to the presence of probiotics. Probiotics contain live bacteria that have the potential to strengthen the intestinal barrier layer as well as modulate the immune system [20]. Hence, we conducted a randomized, double-blind, placebo-controlled trial to assess whether probiotic supplementation can improve clinical biomarkers in NAFLD patients. Since GLA plays an important role in the pathophysiology of NAFLD, it is prudent to explore microbial composition through 16S rRNA sequencing after six-month supplementation with probiotics. We also determined the effect of probiotics on the intestinal mucosal inflammatory cytokines expression of IL-6, TNF-α and IFN-γ. A small intestinal permeability investigation was also carried out by conducting an analysis of protein expression with a focus on the tight junction, zonula occluden-1 (ZO-1) and zonulin.

## 2. Methodology

### 2.1. Study Design

A randomized, double-blind, placebo-controlled study involving patients from Universiti Kebangsaan Malaysia Medical Centre (UKMMC) was conducted. The protocol was approved the by the institutional ethics committee (UKM PPI/111/8/JEP-2019-456). The trial was registered at the US National Institutes of Health website (http://www.clinical-trials.gov, accessed on 30 August 2019) #NCT04074889.

### 2.2. Sample Size Calculation

Sample size was determined using Power and Sample Size (PS) software version 3.1.2. The study design was set as a randomized controlled clinical trial. The determination analysis carried out was an unpaired-*t*-test. The input variable α is the probability of making a type I error. The power of the study is the correct probability of rejecting the null hypothesis and has been set at 80% (0.8). This study is based on a previous study where the reading of hepatic steatosis (CAP) using an ultrasound scan was the main objective of the study [21]. According to the calculations made, a total of 28 samples are required for this study. However, a dropout rate as high as 30% dropout rate is predicted, therefore the number of samples is increased by 12 samples to make a total of 40 samples. For the 16S rRNA amplicon sequencing analysis, 12 samples were required for each group, thus the sample was increased to 48 (24 samples per group).

### 2.3. Patients and Sample Collection

The inclusion criteria were as follows: patients aged 18 years old and above with an ultrasound diagnosis of fatty liver, a baseline-controlled attenuation parameter (CAP) score measured by transient elastography (FibroScan) of ≥263 and baseline alanine aminotransferase (ALT) of more than 35 IU/L for males and 25 IU/L for females. Patients with evidence of other chronic liver diseases such as concomitant hepatitis B or C infections, autoimmune hepatitis disorder or alcoholic liver disease were excluded from this study. Other exclusion criteria consisted of evidence of acute disorders affecting the liver such as drug-induced liver injury, the presence of hepatocellular carcinoma (or liver metastases), any biliary diseases (which would explain the raised ALT, such as gallstones) or evidence of liver cirrhosis. Patients were advised to stop taking any nutritional supplements and to temporarily discontinue any lipid-lowering drugs, beginning at least four weeks prior to the study. The recruitment period lasted for a period of six months (September 2019 to February 2020). All patients provided their written informed consent. At baseline measurement, patients’ comorbidities (diabetes mellitus, hypertension and dyslipidaemia) were recorded.

Body mass index (BMI) classification was performed based on the Malaysian Clinical Practice Guidelines (CPG) 2004 [22]. Individuals with a BMI of around <18.5 kg/m^2^ were classified as being underweight, 18.5–22.9 kg/m^2^ as normal and >23.0 kg/m^2^ as overweight. A BMI score in the range of 27.5–34.9 kg/m^2^ was categorized as obese, 35.0–39.9 kg/m^2^ as obese class two and >40.0 kg/m^2^ as obese class three. During the sampling process, esophagogastroduodenoscopy (OGD) was performed on the NAFLD patient, whereby biopsies (4–5 bites) were obtained from the duodenum part 2 (the descending part). The biopsy tissues were then placed in a cryovial containing 300 μL of RNA later. The tissue was then frozen at −80 °C until it was used. These tissues were used for 16S rRNA amplicon sequencing, immunohistochemistry (IHC) and real-time quantitative polymerase chain reaction (RT-qPCR). For the blood sample, a total of 10 mL was obtained from the patient for two different blood tubes, namely the serum tube and the sodium fluoride tube. These two blood tubes are important for the purpose of blood biochemistry test analysis for ten types of biomarkers. The blood tubes were then stored in a refrigerator at 4 °C for no more than three hours before being processed for further analysis. For ELISA analysis, frozen serum samples that were stored at −80 °C for more than six months were used. Figure 1 shows the CONSORT diagram of this study.

### 2.4. Intervention

The participants were randomised to receive either probiotics or placebo. The probiotics used were HEXBIO^®^ Microbial Cell Preparation (MCP), from B-Crobes Laboratory Sdn. Bhd, which contain MCP^®^ BCMC^®^ strains. Each sachet (3 g) consists of a total of 30 billion colony-forming units (CFU) with six probiotic strains (*Lactobacillus acidophilus* BCMC^®^ 12130 (107 mg), *Lactobacillus casei* subsp. BCMC^®^ 12313 (107 mg), *Lactobacillus lactis* BCMC^®^ 12451 (107 mg), *Bifidobacterium bifidum* BCMC^®^ 02290 (107 mg), *Bifidobacterium infantis* BCMC^®^ 02129 (107 mg) and *Bifidobacterium longum* BCMC^®^ 02120 (107 mg)). Meanwhile, participants in the placebo group received an identical sachet without probiotic strains.

The participants were instructed to consume one sachet twice daily (in the morning and evening, either with or without meals) for 6 months, where they consumed directly or mixed with room temperature water. Sachets were stored in a dry place below 25 °C and away from direct sunlight.

### 2.5. 16S rRNA Sequencing

#### 2.5.1. Illumina Library Generation

DNA was extracted from duodenal biopsy samples using the Ultra-Deep Microbiome Prep (Molzym, Bremen, Germany). The bacterial DNA was amplified by targeting the V3 hypervariable region of 16S rRNA using the primers with partial Illumina adapters based on PRBA338fGC and PRUN518r [23]. PCR was performed using WizBio HotStart PCR mastermix (WizBio, Gyeonggi, Republic of Korea) using the PCR profile of: 95 °C for 3 min followed by 35 cycles of 95 °C for 30 s, 55 °C for 20 s and 72 °C for 20 s [24]. PCR products were purified using SPRI Bead (Beckman Coulter, CA, USA) [25] followed by index PCR to incorporate an Illumina dual index barcode. The barcoded libraries were inspected on gel, pooled according to band intensity and gel-purified using the WizPrep™ Gel/PCR Purification Mini Kit (WizBio, Gyeonggi, Republic of Korea) according to the manufacturer’s instructions. Quantification of the pooled libraries used Denovix high sensitivity and an appropriate amount of the libraries were loaded onto an iSeq100 (Illumina, San Diego, CA, USA) for 2 × 150 paired-end sequencing or 1 × 250 single-end sequencing.

#### 2.5.2. Bioinformatics

Raw paired-end reads were adapter-trimmed and overlapped using fastp v0.21 [26] Forward and reverse primer sequences at the 5′ and 3′ ends of the merged reads, respectively, were removed with cutadapt v1.18 [27]. The merged and primer-trimmed reads were denoised with dada2 [28] within the QIIME2 v.2021.4 [29]. Taxonomic assignment of the Amplicon Sequence Variants (ASVs) used a q2-feature-classifier [30] trained on the latest GTDB release r202 16S rRNA database (trimmed to only retain the V3 hypervariable region) that is comprised of 254,090 bacterial and 4316 archeal genomes organized into 45,555 bacterial and 2339 archaeal species clusters [31]. Both the ASVs table and taxonomic classification table were exported using QIIME2 tools into tab-separated values and manually formatted to generate MicrobiomeAnalystcompatible input [32]. ASVs assigned to only the Kingdom level (Bacteria) were assessed using BLAST against the nt database and most of these ASVs showed a high similarity in identity to the human genome suggesting non-specific amplification of the host (*Homo sapiens*) DNA. As such, the final data output was filtered to ensure that the analysis was conducted only on microbial-derived ASVs.

### 2.6. RT-qPCR

A total of 28 NAFLD and 11 control small intestinal biopsy samples were subjected to RT-qPCR to assess the mRNA expression of inflammatory cytokines, including IL-6, TNF-α and IFN-γ. The primer sequences are summarized in the Appendix A. The expression of each inflammatory cytokines was assessed relative to the housekeeping gene GAPDH. Each sample was evaluated in duplicate. The results were normalized to the expression of the GAPDH gene.

### 2.7. ELISA

#### Serum Zonulin and ZO-1 Concentrations

A total of 40 blood samples were subjected to ELISA analysis to assess the concentration of serum zonulin and ZO-1. After blood samples were collected, the samples were left to clot for one hour at room temperature or overnight at 2–8 °C before centrifugation for 20 min at 1000× *g*. The supernatant was collected to carry out the assays. These assays employed the quantitative sandwich enzyme immunoassay technique. The kit was provided by Elabscience Biotechnology Co., Ltd., Wuhan, China.

Principle of the assay: The ELISA microplates provided in the kit were pre-coated with an antibody specific to human zonulin and zonula occludens-1 (ZO-1). Samples (or standards) were added into the ELISA microplate wells and combined with the specific antibody. Then, a biotinylated detection antibody specific for human zonulin, ZO-1 and Horseradish Peroxidase (HRP) conjugate were added successively to each micro plate well and incubated. Free components were then washed away. Next, the substrate solution was added to each well. Only those wells that contained human zonulin, ZO-1, biotinylated detection antibody and Avidin-HRP conjugate appeared blue in colour. The enzyme-substrate reaction was terminated by the addition of stop solution which caused the colour to change t to yellow. The optical density (OD) was measured spectrophotometrically at a wavelength of 450 nm. The OD value was proportional to the concentration of human zonulin and ZO-1. The detection range of human zonulin is typically between 0.78–50 ng/mL, while for human ZO-1, it is between 0.16–10 ng/mL. The intra- and interassay coefficients of variation for both assays were <10%.

### 2.8. Statistical Analysis

The data were statistically analysed using the GraphPad Prism software version 9. The data were presented as the mean ± s.d. Differences in the gut microbiota at different levels between pre-intervention and post intervention were analysed using the student’s *t*-test. The analysis of variance (ANOVA) was performed to determine the changes of expression of mRNA inflammatory cytokines in both NAFLD and healthy controls.

A compehensive statistical analysis of microbiome data was performed using MicrobiomeAnalyst with the Marker-Gene-Data Profiling (MDP) module [33] for bacterial composition comparison across different groups with adjusted parameters [34]. The low-count features were filtered with a minimum count cut-off of four based on 20% prevalence mean values. Features with a variation of less than 10% were excluded based on the interquartile range and adjusted using cumulative sum scaling (CSS) [35]. The observed species and Shannon indexes were used to assess alpha-diversity, while the Jaccard Index and Bray-Curtis dissimilarity matrices presented in principal coordinates analysis (PCoA) were used to assess beta-diversity. The linear discriminative analysis effect size (LefSe) was used to identify the features with significant abundance between the groups at *p* < 0.05. For ELISA analysis, differences between pre-intervention and post-intervention in both probiotics and placebo groups were determined using a paired-*t*-test analysis. A value of *p* < 0.05 was considered significant.

## 3. Results

### 3.1. Baseline Characteristics

In total, 83% of the patients enrolled in the study completed the intervention, with 18 patients in the probiotic’s groups and 22 patients in the placebo group. The baseline demographics data are summarized in Table 1. Eight patients withdrew from the study due to work commitments, pregnancy and logistics issues. Based on Table 2, there was no significant difference recorded for clinical parameters in both probiotic and placebo groups after a six-month intervention.

### 3.2. 16S rRNA Sequencing

We investigated 58 out of 64 biopsy samples of NAFLD patients for gut microbiome analysis based on the V3 region via amplicon. Due to low sequencing read outputs, six samples did not pass the quality control (QC) and were removed from this investigation. After applying strict trimming criteria to exclude low-quality reads and human genomic artifacts, a total of 920,434 reads were obtained with 3786 ASVs. On average, 57,039 reads were generated per sample (min: 16,895; max: 132,586). At the phylum level, five predominant phyla were found in both probiotics and placebo groups which include Actinobacteria, Proteobacteria, Firmicutes, Bacteroidota and Fusobacteria. The probiotics group experienced an increase in the Actinobacteria phylum after a six-month intervention, whereas placebo groups had a decrease in this phylum. We conducted an alpha-diversity analysis for both treatment groups to investigate the richness and evenness of microbial communities in a community species (Figure 2). In comparison to baseline data, the probiotics groups showed a decrease in the richness of microbiota (−2.75 ± 6.17; 95% CI; *p* = 0.156), while the placebo group showed an increase after a six-month intervention (0.57 ± 5.97; 95% CI; *p* = 0.73). Figure 2 shows that there was no significant difference in microbiota biodiversity between the probiotic and placebo groups (*F* value [ANOVA] = 0.51, *p* = 0.67).

The Bray-Curtis dissimilarity analysis and the Jaccard Indexes were used to calculate the diversity between groups. The PCoA based on the Bray-Curtis dissimilarities index showed significant differences between the species community after the intervention in the probiotics group, as shown in Figure 3A,B. Based on the PERMANOVA value, the probiotics group formed a significant formation of clusters compared to the placebo group (probiotic, *p* = 0.003 and placebo, *p* = 0.054). The same results were obtained for the Jaccard index analysis, where the probiotics group also showed a significant similarity value compared to the placebo group (*p* < 0.05).

Using LefSe, five unclassifiable species of gut microbiota were discovered for the probiotics group. *Unclassified_Fusobacterium*, *unclassified_Clostridia*, *unclassified_Strenotrophomonas, unclassified_Streptococcus*, and *unclassified_Proteobacteria* were among the species discovered. In the placebo group, on the other hand, we found six dominant species: *unclassified_neisseria, unclassified_Crytophagales*, *Rothia mucilaginosa, Prevotella melaninogenica, unclassified_Rothia*, and *unclassified_Escherichia* (Figure 4). The probiotics group experienced a significant decrease in *unclassified_Proteobacteria, unclassified_Streptococcus*, and *unclassified_Strenotrophomonas* species compared to before the intervention. *Unclassified_Fusobacterium* and *unclassified_Clostridium* species, on the other hand, showed a significant increase in the same treatment group. Meanwhile, the placebo group experienced a significant increase for the following four species: *unclassified_Neisseria, Rothia mucilaginosa, Prevotella melaninogenica*, and *unclassified_Rothia.*

### 3.3. Expression of Small Intestinal Inflammatory Cytokines in Healthy and NAFLD Group

#### 3.3.1. Relative Expression of IFN-γ in Duodenal Tissue

The probiotics group demonstrated a 7.9-fold decreased after six months of intervention period (from 8.835 ± 1.1 to 0.9781 ± 0.13; *p* < 0.001). There was no significant difference in the post-relative expression of IFN-γ in the placebo group compared to pre-intervention (from 7.463 ± 3.18 to 4.049 ± 0.92; *p* = 0.16) (Figure 5).

#### 3.3.2. Relative Expression of TNF-α in Duodenal Tissue

The probiotics group showed a significant downward trend compared to before the intervention (from 2.432 ± 0.28 to 1.469 ± 0.19; *p* = 0.003). This group also displayed a significant difference in the findings of post-intervention compared to the control group (*p* < 0.05). Meanwhile, the placebo group showed no significant difference in the post-relative expression of TNF-α compared to pre-intervention (from 1.440 ± 0.31 to 1.908 ± 0.51; *p* = 0.78) (Figure 6).

#### 3.3.3. Relative Expression of IL-6 in Duodenal Tissue

The probiotics group exhibited a significant increase in IL-6 expression after the six-month intervention period (from 1.727 ± 0.51 to 14.52 ± 2.06; *p* < 0.001). This group also showed a significant difference between post-intervention expression compared to the control group (+13.52 ± 1.789; *p* < 0.001). The placebo group showed no significant difference between pre- and post-intervention of IL-6 expression (from 2.027 ± 0.68 to 4.432 ± 1.36; *p* = 0.23). However, there was a significant difference between the post-intervention expression and control group expression (+3.342 ± 1.196; *p* = 0.02) (Figure 7).

### 3.4. Expression of Tight Junction and Inflammatory Biomarker Protein in NAFLD Group

#### 3.4.1. Expression of Zonula-Occluden-1 (ZO-1) Protein in Serum Samples

The probiotics group showed a significant decrease in ZO-1 protein expression (from 53.25 ng/mL ± 2.70 to 20.73 ng/mL ± 3.55; *p* < 0.0001). A significant decrease was also observed in the placebo group (from 48.89 ± ng/mL 4.79 to 13.56 ng/mL ± 1.88; *p* < 0.0001) (Figure 8).

#### 3.4.2. Expression of Zonulin Protein in Serum Samples

The probiotics group showed a significant increase after the six-month intervention period (from 119.4 ng/mL ± 15.26 to 278.3 ng/mL ± 42.6; *p* < 0.01). The same result was obtained when the placebo group also recorded for a significant increase for post-intervention compared to pre-intervention (from 181.7 ng/mL ± 25.69 to 281.5 ng/mL ± 41.50; *p* < 0.01) (Figure 9).

## 4. Discussion

Currently, the prevalence of NAFLD remains alarmingly high worldwide. Since the inception of GLA’s concept, mounting studies suggest the potential role of gut microbiota during intestinal dysbiosis, which is associated with the pathophysiology of NAFLD [36]. While the treatments for NAFLD have been limited to focusing on a healthy diet and active lifestyle modifications, probiotic supplementation may improve the risk factors of NAFLD by affecting the intestinal microbiota. As a result, we reported the first stratified randomised study in Malaysia that focuses on microbiota manipulation via probiotic supplementation as one of the additional treatments for NAFLD.

The compound annual growth rate (CAGR) of global demand for probiotics increased by 8.08% between 2022 and 2027 [37]. This increase demonstrates that the general public is becoming more aware of the importance of maintaining a healthy digestive system. HEXBIO^®^ (MCP^®^ BCMC^®^ strains) was chosen for this study because it contains six viable probiotics from *Bifidobacterium* and *Lactobacillus* strains. Members of the Bifidobacterium and Lactobacillus genera contribute significantly to healthy gut microbiota. Bifidobacterium is recognised as the dominant bacterial genus present in healthy infants’ gut [38], whereas Lactobacillus has been known to contain a high number of GRAS (generally recognised as safe) species by the United States Food and Drug Administration (USFDA) [39,40]. These multi-strain probiotics contain 30 billion CFU, which have been shown to improve gastrointestinal health [41,42,43], improve diabetes management [44,45,46], strengthen the immune system [47,48] and speed up recovery from hospitalisation and ICU [49,50] through administration in clinical trials. *Lactobacillus acidophilus* and *Lactobacillus paracasei* strains were discovered to improve stool consistency and reduce strain symptoms in IBS patients suffering from chronic constipation [51]. Other studies by our research group also found that after four weeks of taking probiotics (MCP^®^ BCMC^®^ strains), the secretion of pro-inflammatory cytokines decreased in colorectal cancer patients who had undergone surgery [47]. Biomarkers for biochemical tests such as haemoglobin A1C (HbA1c) and insulin decreased significantly in T2DM patients after 12 and 24 weeks of probiotic intervention [44,52].

There was no statistically significant improvement in any biochemical blood parameters that have been found in our study. To the best of our knowledge, this is the first study to be conducted in a Malaysian setting that focuses on microbiota manipulation in the small intestinal of NAFLD patients. A randomised controlled trial of VSL#3^®^ probiotics failed to show any reduction in biochemical tests such as ALT, SST, cholesterol, and others, implying that this strain did not improve liver damage [32]. The implementation of the symbiotic mixture containing *Bifidobacterium animalis subspecies lactis* (*BB12*) and prebiotic Actilight^®^ 950P for 10 to 14 months found no significant decrease in the level of hepatic steatosis in 106 NAFLD patients [53]. Several important indicators, such as strain type, treatment dose, average patient age, sample size, gender, intervention period, study location, dietary patterns, and the population involved, should be considered [54]. A previous study found that prebiotics containing oligofructose are more effective than probiotics in lowering the fat profile [55]. For liver enzyme indicators, however, there is no difference between probiotics and prebiotics. This is because the effect of probiotics on liver enzymes such as ALT and AST vary [56]. The systematic review also found that probiotics are more effective in studies with fewer than 300 participants and a study period of more than 16 weeks.

Currently, haemoglobin A1c (HbA1c) is used as one of the biomarkers to assess the level of glycaemic control in NAFLD patients [57]. In both diabetic and non-diabetic patients, the level of HbA1c in the blood correlates with the severity of hyperglycaemia [58]. After two months, a randomized, blinded clinical trial that examined the effectiveness of various strains of probiotics reported a significant decrease in glycaemic levels as well as the expression of inflammation-related proteins in NAFLD patients [59]. Our study, on the other hand, used fasting blood sugar (FBS) biomarkers rather than HbA1c for blood biochemical analysis. This is due to the need for a fasting blood sugar (FBS) biomarker for the next analysis, Liverfast.

16S rRNA sequencing of small intestinal biopsy samples revealed the following five predominant phyla in both the probiotic and placebo groups: Actinobacteria, Proteobacteria, Firmicutes, Bacteroidota and Fusobacteria. The probiotics group experienced an increase in the Actinobacteria phylum and a decrease in the Proteobacteria phylum after a six-month intervention period. Meanwhile, the phylum Actinobacteria level decreased while that of the phylum Proteobacteria increased in the placebo group. This study’s findings are consistent with the findings of a recent study, which revealed an increase in the phylum Proteobacteria in the NAFLD group and Actinobacteria in the healthy control group [60]. The researchers collected colonic mucosal biopsy samples from 20 NAFLD patients and 20 healthy people as controls. The same result was obtained in a stool sample study, where the Actinobacteria phylum was found to be significantly lower in the NASH and obese NAFLD groups [61,62].

Proteobacteria is used as one of the benchmarks for the microbiota population that contributes to the formation of a disease caused by inflammation [63]. An increase in the phylum Proteobacteria was discovered in a mouse model fed a high-fructose diet, which is attributed to an increase in LPS moving into the liver via the blood vessels [64]. Interestingly, this phylum is also used to determine the degree of fibrosis in NAFLD patients [65]. The Actinobacteria phylum, particularly the *Bifidobacteria* family, has received attention because it can benefit the intestine in terms of the immune system, the intestinal border, or metabolism [66]. The presence of MCP^®^ BCMC^®^ strains of *Bifidobacterium bifidum, Bifidobacterium infantis*, and *Bifidobacterium longum* in the probiotics used in this study is likely to be responsible for the increase in the Actinobacteria phylum in the probiotics group.

In particular, the relative abundance values obtained in this study are consistent with the findings of the LefSe. After six months, the probiotics group was able to significantly reduce *unclassified_Proteobacteria* and *unclassified_Streptococcus* species. In contrast, the placebo group experienced a significant increase in pathogenic bacteria such as *Rothia mucilaginosa* and *Prevotella melaninogenica*. A study discovered the *Rothia mucilaginosa* species in both isolated systolic hypertension (ISH) and isolated diastolic hypertension (IDH) groups. In 2021, a significant decrease in this species was found in the metabolism activator (CMA) treatment group for NAFLD patients after 70 days [67]. In general, this species produces acetaldehyde, which has the potential to cause DNA damage and mutagenesis [68]. *Prevotella melaninogenica* is commonly found in periodontium patients, which correlates with NAFLD patients’ liver fibrosis levels [69]. While alpha-diversity, which represents species richness and evenness, showed no significant difference between pre- and post-intervention in both groups, the beta-diversity analysis showed a significant difference in the probiotics group. The findings of this study show that taking probiotics significantly modifies the composition of the microbiota via cluster formation. Sample size influences alpha-diversity analysis [70]. Only 58 of the 64 samples in this study were successfully sequenced. Six samples were excluded from the analysis due to a low yield of sequencing results.

Butyrate is the primary nutrient source for colonocytes. Clostridia and Bacteroidetes are the most common butyrate-producing bacteria found in anaerobic conditions with low oxygen concentrations. Members of Actinobacteria, Fusobacteria, Spirochaetes, and Proteobacteria, in contrast hand, have a high potential as butyrate producers based on the genes they express [71]. Butyrate helped contribute to intestinal barrier integrity by altering tight junction proteins and mucin [72]. As a matter of fact, this group of bacteria may be able to reduce systemic inflammation by restoring gut permeability. After six months of probiotic supplementation, we found a significant increase in *unclassified_Clostridium* and *unclassified_Fusobacterium*. These data are consistent with findings from a systematic review, which found that the phyla Fusobacteria, Lentisphaerae, Proteobacteria, Thermus, and Verrucomicrobia were lower in NAFLD patients compared to the control group [73].

In each part of the gastrointestinal (GI) tract, the human microbiota differs both taxonomically and functionally [74]. The number of microbiotas in the stomach is low, while it is high in the colon [75]. This is consistent with the composition of the microbiota, which is also distinct, with phylum Firmicutes and Bacteroidetes predominating in the lower gastrointestinal tract and the phylum Proteobacteria and Firmicutes predominating in the upper gastrointestinal tract [76]. The role of microbiota in the pathogenesis of NAFLD begins with the translocation of bacterial products into the liver when intestinal permeability increases [77]. This situation is dependent on the interaction of the GLA with the innate immune response.

Many studies have been conducted in recent years to investigate the composition of microbiota found in NAFLD patients using faecal samples, focusing on V3 and V4 regions. The most common bacteria composition that is found in the patients includes Firmicutes, Proteobacteria and Bacteroidetes [78,79,80]. Previous studies revealed the presence of *Escherichia_Shigella and Lachnospiraceae_Incertae_Sedis* based on a 16S rDNA sequencing analysis in NAFLD patients [81] and NASH in animal model [82]. The non-invasive faecal sampling method is convenient, but it has the potential to reveal that an uneven distribution of bacteria that was affected during homogenization processes [83]. Biopsy, on the other hand, is more precise in investigating the association of microbiota within the targeted tissue. The GLA is defined as a close interaction between the liver and the gastrointestinal tract that includes microbiota and the immune system response [5,84]. Small intestinal bacterial overgrowth (SIBO), dysbiosis, and increased intestinal permeability, also known as ‘leaky gut’, are among the changes. However, according to the Malaysian perspective, the use of biopsy that focuses on small intestinal areas in NAFLD patients is currently limited.

After the six-month intervention, IFN-γ and TNF-α gene expression levels decreased significantly in the probiotics group. The findings of this study are consistent with other studies that found a decrease in the expression of the same genes in the control group compared to NAFLD patients [85]. The presence of probiotic strains that can improve tight junctions may influence the gut microbiota. Most previous studies studied the expression level of cytokines related to inflammation using plasma or serum samples rather than biopsies and found the same results as this study [86,87,88]. IL-6 is a pleiotropic cytokine that plays various physiological functions in a specific cell or tissue [89]. Through lymphocyte proliferation, IL-6 is commonly used as a marker of inflammation [90]. Nevertheless, IL-6 has the potential to act as a pro-inflammatory or anti-inflammatory cytokine through different transcriptional regulation mechanisms [91].

This study discovered that after six months, the probiotics group had a significant increase in the IL-6 gene. The findings of this study are consistent with an in vivo study that successfully improved the intestinal permeability of a mouse model by finding the Bacteroidales order, which influences the expression of IL-6 [92]. This finding is consistent with the phylum Bacteroidota’s increase in relative abundance values obtained in this study through 16S rRNA sequencing analysis. Another in vivo study using the *Pediococcus* acidilactici K15 strain found that it could influence IL-6 production by producing immunoglobulin A (IgA) [93]. In addition, an increase in IL-6 levels was also associated with toll-2 receptor function, which influences IgA production [94]. The researchers obtained their findings by using strains of *L. casei* CRL 431 and *L. helveticus* R389 in a mouse model for two to seven days. The liver-intestinal axis in NAFLD pathophysiology influences the mucosal immune trigger. This situation can be controlled through the presence of tight junctions in the intestinal epithelial permeability layer [95,96]. Zonulin, a novel protein that was discovered in 2000, acts as a biomarker for tight junction damage in intestinal structures [97]. Many recent studies found increased zonulin expression in NAFLD patients [98,99,100]. Further investigation of tight junctions and the presence of zonulin may allow researchers to gain a better understanding of the pathophysiology of NAFLD.

In our recent publication, the expression of the tight junction protein zonula occluden-1 (ZO-1) decreased significantly in the placebo group using the IHC technique on duodenal biopsy tissue samples [101]. This study’s findings are consistent with those of previous research that found a decrease in ZO-1 protein expression in the NAFLD group compared to the control group [102]. A decrease in the expression of ZO-1 and ZO-2 was reported in a high-fat-fed mouse model [103]. According to the study, supplementing the control group of mice models with probiotics containing a combination of Bifidobacteria, Lactobacilli, and Streptococcus did not result in any significant changes. The decrease in ZO-1 expression in the placebo group in this study could be associated with increased pathogen and endotoxin translocation through the intestinal epithelial layer. The probiotics group showed a non-significant increase in ZO-1 expression in this study. To more accurately assess the strain’s efficiency, an increase in the intervention period for the treatment group is proposed. ELISA analysis in this study, however, revealed a significant decrease in ZO-1 expression in the probiotics group. In contrast, zonulin protein expression increased significantly after the intervention period in the same group. The findings of this study contradict other previous studies that found a decrease in ZO-1 expression and an increase in zonulin expression in NAFLD patients compared to the control group [98,100,104].

Protein isolation methods for samples derived from circulation in the body, such as serum and plasma, are still being debated [105]. Seeing as serum samples contain a greater number of metabolites than plasma, they may provide more accurate research findings. In this study, frozen serum samples stored in a freezer at −80 °C for more than six months were used for ELISA. The storage time and temperature were demonstrated to have a significant impact on the stability of serum samples [106]. In the study, albumin concentration was measured using serum samples stored at −80 °C for a month and found to be significantly different. Furthermore, frozen samples were demonstrated to have the potential to influence several stresses, such as protein denaturation and pH changes caused by solution crystallisation [107]. The aggregation of folded proteins has the potential to cause protein structural instability [108].

Circulating proteins can experience concentration changes as a result of aging, inflammation, or disease formation [109]. Previous research has shown that when the circulating protein alpha-2 macroglobulin is bound with inflammatory cytokines such as IL-4 and IL-10, the expression rate of the protein changes [110]. Protein quantification, as opposed to IHC techniques, focuses on localised proteins in biopsy tissue samples [111]. Nonetheless, a study that compared ELISA and IHC techniques to detect the presence of plasminogen system activation components in human tumour tissue concluded that these two techniques do not have a clear correlation, therefore different study results are probable [112].

However, some limitations of the study have been identified and can be improved on in future research. The first limitation of this study is the small sample size for the treatment group. A larger sample size will aid in obtaining more accurate study results. Indeed, ethnic diversity, inclusion criteria, data collection methods for patient nutrition, patient compliance rates with probiotic intake, and statistical analysis may all have an impact on the study’s findings. Next, the role of probiotics in this study may not be representative of the Malaysian population because the trial was conducted at a single health centre where the majority of participants were of Malay ethnicity. The effectiveness of probiotics is also affected by the presence of a variety of strains and the duration of the intervention used in a study [113]. Following the presence of the COVID-19 pandemic, we did not have sufficient time to examine the differences in study findings after the consumption of probiotics was discontinued, also known as the “washout period.” Following the implementation of the movement control order (MCO) in 2021, some patients from this study experienced interruptions in receiving probiotics for the second phase of the study. As a result, probiotic intake in patients may be less effective. This study also used serum samples that had been frozen at −80 °C. This potentially affects the efficacy of probiotic intake in patients. For ELISA purposes, this study also used serum samples that were stored in a freezer at −80 °C for more than six months. This issue is likely to result in different results from previous studies due to the influence of protein denaturation in the sample.

## 5. Conclusions

Overall, probiotics that contain six viable species derived from *Bifidobacterium* and *Lactobacillus* strains improved the gut microenvironment by ameliorating gut microbiota imbalance via a significant change in beta-diversity analysis. Pathogenic bacteria were found to significantly increase after the six-month intervention when *Lactobacillus* and *Bifidobacterium* species contained in the probiotics were absent. The reversal of dysbiosis in the probiotics group reduced inflammatory cytokine secretion, which suggested a lower translocation of bacterial toxin across the GLA. Although our study did not show any positive changes in small intestinal permeability, our previous preliminary study on the same protein demonstrated that the small intestinal barrier can be restored. As a result, probiotics should be considered as an adjunct alongside a balanced diet and the implementation of a healthy lifestyle.

## Figures and Tables

**Figure 1 biomedicines-11-00640-f001:**
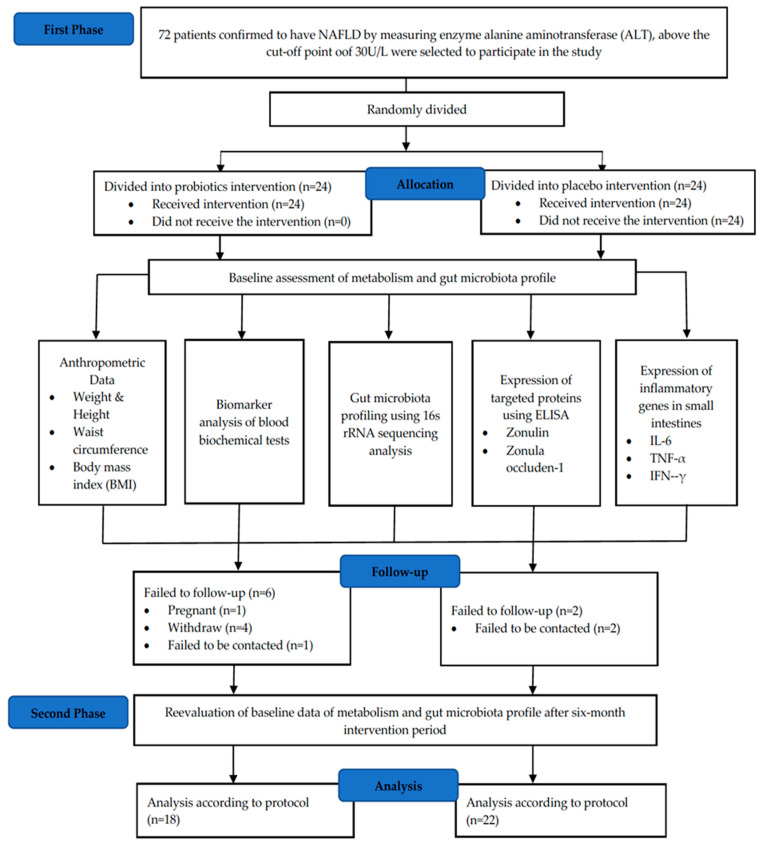
CONSORT diagram of randomized double-blind controlled clinical trials for a period of six months., enzyme-linked immunosorbent assay; TNF-α, tumour necrosis factor-alpha; IL-6, interleukin-6; IFN-γ, interferon-gamma; rRNA, ribosomal RNA.

**Figure 2 biomedicines-11-00640-f002:**
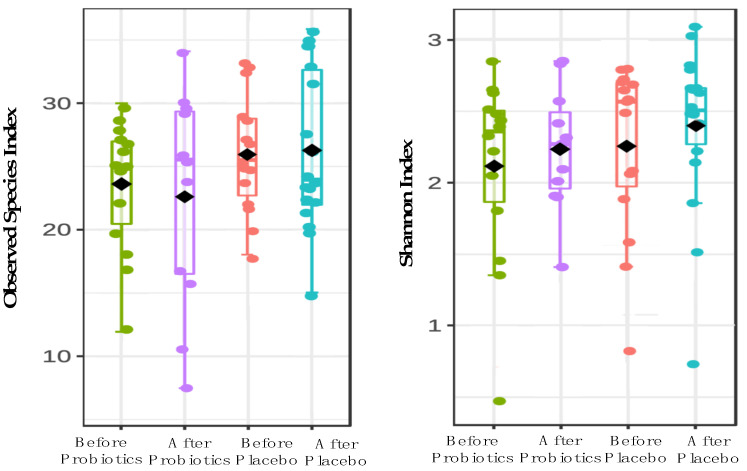
The alpha-diversity analysis using Observed species and Shannon indexes at genus level for both probiotics and placebo groups.

**Figure 3 biomedicines-11-00640-f003:**
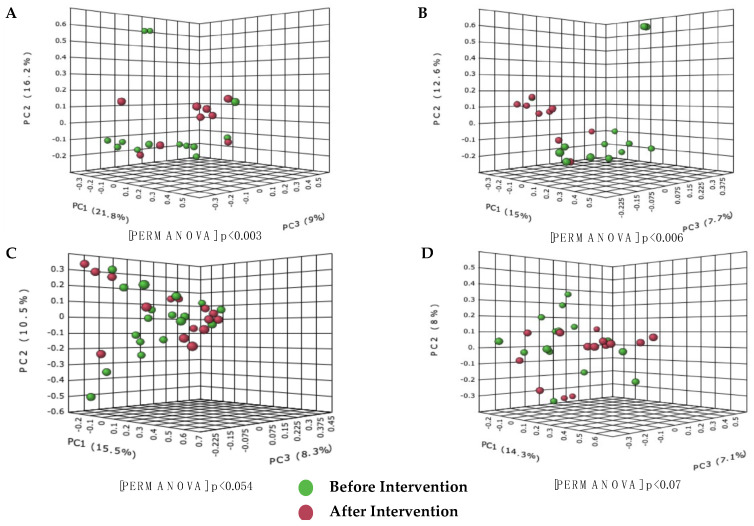
Cluster analysis showing differences in the species composition of gut microbiota present before and after the probiotic and placebo intervention in the treatment group. (**A**) Bray−Curtis dissimilarity analysis before and after intervention in the probiotics group; (**B**) Jaccard index before and after the probiotics group intervention; (**C**) Bray−Curtis dissimilarity analysis before and after the intervention in the placebo group; (**D**) Jaccard index before and after the intervention in placebo group. Differences between groups were defined as significant for a value of *p* < 0.05.

**Figure 4 biomedicines-11-00640-f004:**
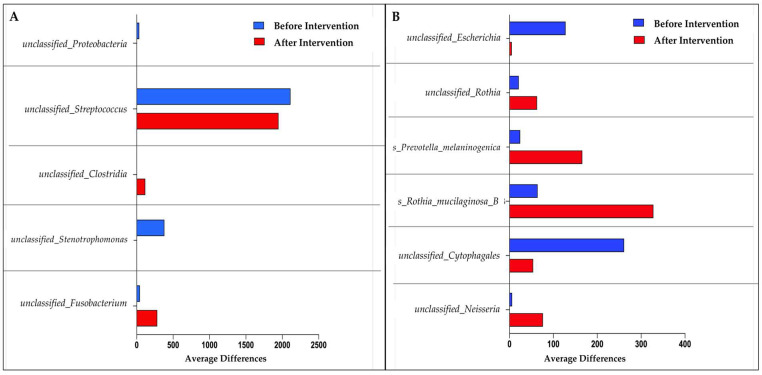
Bar graph based on LefSe analysis that demonstrates the identification of significant intestinal microbiota abundance at the species level. (**A**) Species mean difference before and after probiotic intervention; (**B**) species mean difference before and after placebo intervention.

**Figure 5 biomedicines-11-00640-f005:**
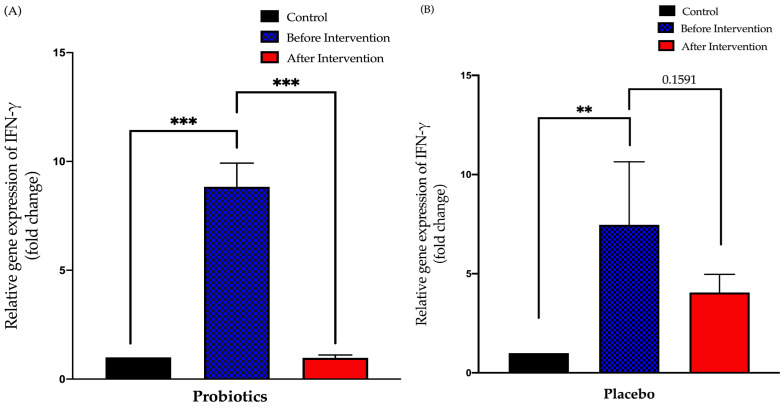
Bar graph showing the relative expression of the IFN-γ gene (fold change) in the duodenal tissue of NAFLD patients. (**A**) Relative expression of IFN-γ gene (fold change) in the probiotics group; (**B**) relative expression of IFN-γ gene (fold change) in the placebo group. Values are expressed as mean ± SEM (** *p* < 0.01 and *** *p* < 0.0001). The probiotics group showed a significant decrease compared to before the intervention while the placebo group did not show any significant change after six months.

**Figure 6 biomedicines-11-00640-f006:**
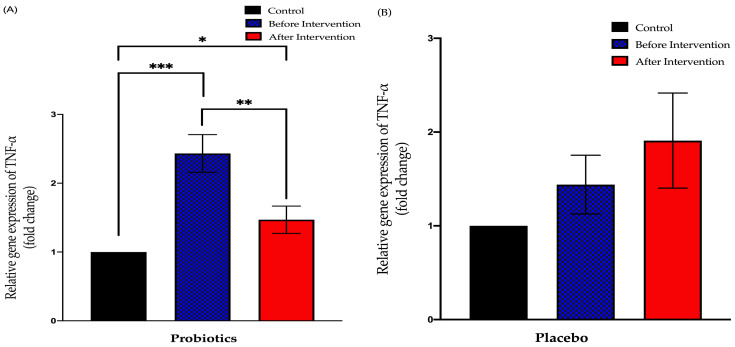
Bar graph showing relative expression of TNF-α gene (fold change) in duodenal tissue of NAFLD patients. (**A**) Relative expression of TNF-α gene (fold change) in the probiotics group; (**B**) relative expression of TNF-α gene (fold change) in the placebo group. Values are expressed as mean ± SEM (* *p* < 0.05, ** *p* < 0.01 and *** *p* < 0.0001). The probiotics group showed a significant downward trend compared to before the intervention while the placebo group did not show any significant change after the six-month period.

**Figure 7 biomedicines-11-00640-f007:**
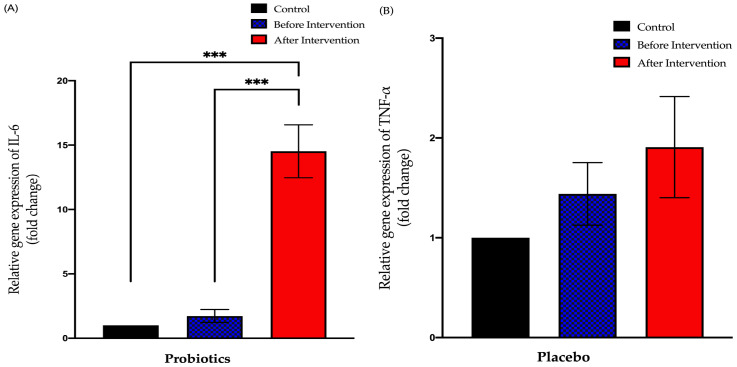
Bar graph showing relative expression of IL-6 gene (fold change) in duodenal tissue of NAFLD patients. (**A**) Relative expression of IL-6 gene (fold change) in the probiotics group; (**B**) relative expression of IL-6 gene (fold change) in the placebo group. Values are expressed as mean ± SEM (*** *p* < 0.0001). The probiotics group showed a significant upward trend compared to before the intervention while the placebo group did not show any significant change after the six-month period.

**Figure 8 biomedicines-11-00640-f008:**
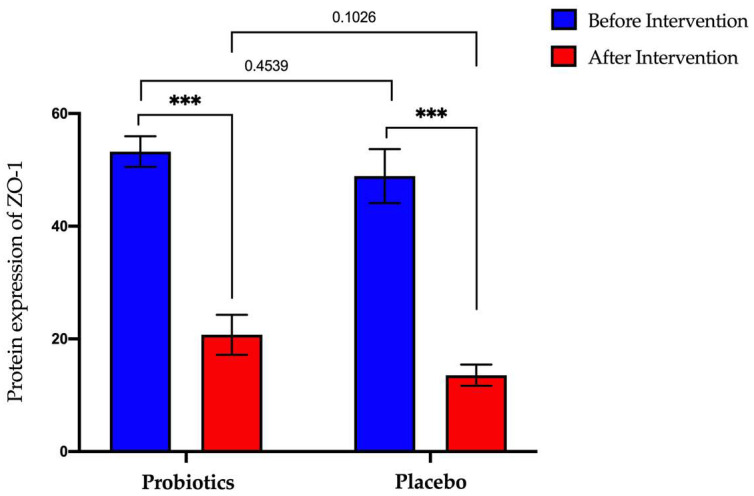
Bar graph showing ZO-1 protein expression before and after intervention in serum samples of NAFLD patients. *p*-values were calculated using a paired *t*-test. Values show mean ± SEM (*** *p* < 0.0001).

**Figure 9 biomedicines-11-00640-f009:**
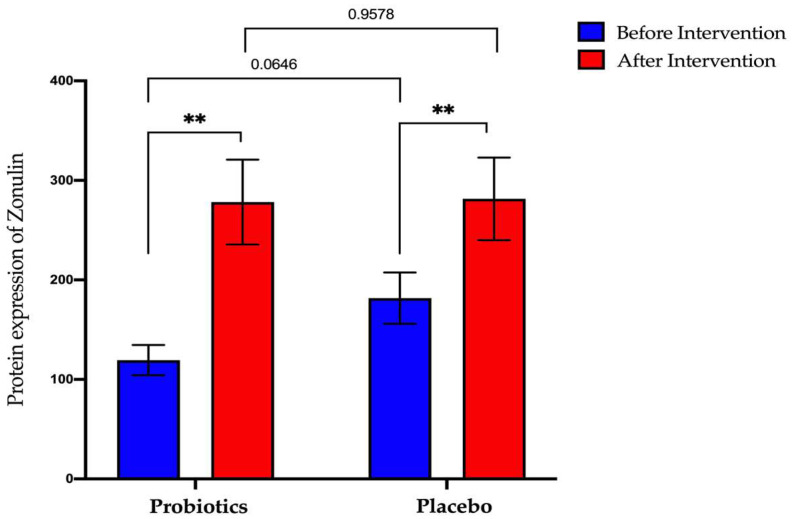
Bar graph showing zonulin protein expression before and after intervention in serum samples of NAFLD patients. *p*-values were calculated using a paired *t*-test. Values show mean ± SEM (** *p* < 0.01).

**Table 1 biomedicines-11-00640-t001:** Demographic analysis of studied patients at baseline.

Characteristics	Total (*n* = 40)	Probiotics (*n* = 18)	Placebo (*n* = 22)	*p*-Value
Age (years)	52.23 ± 12.89	55.00 ± 11.07	49.95 ± 14.05	0.22
Gender, *n* (%)				
Men	29 (72.5)	12 (66.7)	17 (77.3)	0.50
Women	11 (27.5)	6 (33.3)	5 (22.7)	
Ethnicity, *n* (%)				
Malay	25 (65.2)	12 (66.7)	13 (59.1)	0.79
Chinese	11 (27.5)	4 (22.2)	7 (31.8)	
Indian	4 (10.0)	2 (11.1)	2 (9.1)	

Values are presented as the mean (standard deviation), or *n* (%). *p*-value was obtained by using independent-*t*-test. *p*-value < 0.05.

**Table 2 biomedicines-11-00640-t002:** Clinical parameters at baseline and the end of the study by intervention.

	Probiotics (*n* = 18)		*p*-Value	Placebo (*n* = 22)		*p*-Value
	Baseline	End of Study		Baseline	End of Study	
Body mass index, kg/m^2^	28.25 ± 4.36	28.01 ± 4.08	0.68	28.29 ± 3.91	29.19 ± 5.17	0.14
Waist circumference, cm	95.86 ± 9.74	94.61 ± 10.05	0.25	97.84 ± 11.45	97.61 ± 11.11	0.88
ALT (IU/L)	78.67 ± 52.45	86.17 ± 75.02	0.54	78.45 ± 41.18	78.32 ± 43.89	0.98
AST (IU/L)	47.61 ± 16.82	46.28 ± 23.37	0.77	50.59 ± 21.64	46.86 ± 26.77	0.31
GGT (IU/L)	21.90 ± 13.33	17.90 ± 124.70	0.96	19.50 ± 205.9	16.90 ± 207.4	0.84
Triglycerides (mmol/L)	2.10 ± 0.85	2.07 ± 0.98	0.57	2.11 ± 0.77	2.42 ± 1.47	0.34
Total cholesterol (mmol/L)	5.91 ± 0.92	5.62 ± 0.92	<0.0001	5.63 ± 0.92	5.72 ± 1.46	0.65
Glucose (mmol/L)	5.44 ± 1.40	5.76 ± 1.21	0.22	5.61 ± 1.53	5.10 ± 1.53	0.05
Bilirubin (mmol/L)	17.81 ± 8.31	16.91 ± 9.39	0.37	16.46 ± 6.23	17.28 ± 11.28	0.45
Alpha-2-macroglobulin (mmol/L)	1.57 ± 0.46	1.53 ± 0.43	0.37	1.80 ± 0.68	1.81 ± 0.68	0.82
Apolipoprotein (mmol/L)	1.52 ± 0.25	1.41 ± 0.38	0.22	1.50 ± 0.23	1.30 ± 0.38	0.0019
Fasting glucose, mg/dL	5.13 (0.96)	5.6 (1.09)	0.06	5.50 (1.53)	5.14 (0.68)	0.28

ALT, alanine aminotransferase; AST, aspartate aminotransferase, GGT, gamma-glutamyl transferase. Values are presented as the mean (standard deviation). *p* < 0.001.

## Data Availability

The raw data of the clinical outcomes, 16S rRNA sequencing, RT-qPCR and ELISA results are available from the corresponding author on reasonable request.

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
