# Peer review of "The Effects of Probiotics on Small Intestinal Microbiota Composition, Inflammatory Cytokines and Intestinal Permeability in Patients with Non-Alcoholic Fatty Liver Disease"

_biomedicines, 2023, doi:10.3390/biomedicines11020640_

Round 1

Reviewer 1 Report

Ayob et al. investigated the effects of multi-strain probiotics, including Lactobacillus and Bifidobacterium species, on small intestinal gut microbiota, intestinal inflammatory cytokines, and also intestinal permeability in NAFLD patients. They found those probiotics were able to reverse dysbiosis in NAFLD patients, which could provide new insights for the prevention of NAFLD. I only have some minor comments.

1. LEfSe (Linear discriminant analysis Effect Size) was used to determine the significant biomarkers in each group, it would be better to include this analysis.

2. ASV should be presented with the full name when the first time used in the manuscript. Amplicon sequence variant

3. Revise ‘RT-PCR’ to ‘RT-qPCR”

4. I cannot see figure 3.

5. Format should be revised based on the guideline. There are many errors. Please carefully check.

Author Response

We would like to express our heartfelt appreciation for reviewing our manuscript. We have taken your comments and suggestions into account and made a few changes as a result. The response based on your comments is provided below.

1. LEfSe (Linear discriminant analysis Effect Size) was used to determine the significant biomarkers in each group, it would be better to include this analysis.

Response: Bar Graph in Figure 4 is built based on LefSe analysis.(Line 332)

2. ASV should be presented with the full name when the first time used in the manuscript. Amplicon sequence variant.

Response: The amendment has been done in the text (Line 207).

3. Revise ‘RT-PCR’ to ‘RT-qPCR”.

Response: The amendment has been done in the text (Line 147, Line 219, Line 221, Line 679).

4. I cannot see figure 3.

Response: The size of figure has been amended (Line 313).

5. Format should be revised based on the guideline. There are many errors. Please carefully check.  Reference citation numbers should be placed in square brackets, i.e. [ ].

Response: Reference style has been amend according to the journal’s guidelines. Each section has been amended to corresponding style, which can be found in the “Styles” menu of Word

Reviewer 2 Report

Nurainina Ayob and colleagues have published their manuscript titled "The Effect of Probiotics on Small Intestinal Microbiota Composition, Inflammatory Cytokines, and Intestinal Permeability in Patients with Non-Alcoholic Fatty Liver Disease." According to them, the incidence of non-alcoholic fatty liver disease (NAFLD) has increased as metabolic diseases and obesity have increased. NAFLD is primarily caused by sedentary lifestyles and unbalanced diets. Until recently, no medications were approved to treat NAFLD. Consequently, many pharmacological studies have been conducted, including those involving probiotics. NAFLD pathophysiology has been increasingly understood to involve the gut-liver axis (GLA). In order to unravel the role of GLA, we focused on the upper gut area, specifically the duodenum. A multi-strain probiotic supplement containing six different Lactobacillus and Bifidobacterium species was studied in NAFLD patients to determine the effect of 30 billion CFU of MCP® BCMC® strains on small intestinal microbiota, intestinal inflammatory cytokines, and intestinal permeability. In either group, biochemical blood analysis did not reveal any discernible changes after six months of supplementation. In both groups, 16s rRNA sequencing of small intestine biopsies revealed five dominant phyla: Actinobacteria, Proteobacteria, Firmicutes, Bacteroidota, and Fusobacteria. After comparing pre- and post-intervention microbiota compositions, the probiotics group showed a significant change. As measured by quantitative PCR, intestinal mucosal inflammatory cytokines were significantly lower after probiotic administration in comparison to pre-intervention. Conversely, IL-6 showed a reverse pattern, showing a significant increase in expression in the probiotic group. ELISA analysis revealed a significant increase in circulating zonulin in both groups. On the other hand, both groups showed a significant decrease in the expression of zonula occluden-1 (ZO-1). Probiotics (MCP® BCMC® strains) used in this study were generally effective at changing dysbiosis in NAFLD patients according to assessments of the gut microenvironment. This resulted in the reduction of microbial endotoxin translocation through the gut-liver axis, and the stabilization of inflammatory cytokine expression in the gut. The importance of probiotic supplementation as an adjunct to a healthy diet and lifestyle is urgently needed in order to address NAFLD and its underlying causes. Regarding the present manuscript, I would like to make a few comments.

-The reference style should be revised in accordance with the journal's guidelines

-You do not need to explain the abbreviation more than once

-There is a need for a better introduction to facilitate the reading of your manuscript, to explain why the study was proposed as well as its main objective

-Thanks to the authors for providing an explanation of the materials and methods. Here my main concern is how the sample size is calculated.

-Authors are not required to pay for the use of color figures

-There may be a need for an exploratory analysis of the total information that the authors have so that the authors can create a general picture of the situation, such as a PCA or Pearson correlation of the major variables

-I think the discussion is too speculative. Analyze your own results to determine whether they are relevant or not.

Author Response

We would like to express our heartfelt appreciation for reviewing our manuscript. We have taken your comments and suggestions into account and made a few changes as a result. The response based on your comments is provided below.

1. The reference style should be revised in accordance with the journal's guidelines.  Reference citation numbers should be placed in square brackets, i.e. [ ].

Response:  Reference style has been amend according to the journal’s guidelines. Each section has been amended to corresponding style, which can be found in the “Styles” menu of Word.

2. You do not need to explain the abbreviation more than once.

Response:  The amendment has been done in the whole text [GLA, ASV].

3. There is a need for a better introduction to facilitate the reading of your manuscript, to explain why the study was proposed as well as its main objective.

Response:  The amendment has been done in subsection 1 (Line 49-Line 98).

4. Thanks to the authors for providing an explanation of the materials and methods. Here my main concern is how the sample size is calculated.

Response:  The amendment has been added in the text (Line 109).

5. Authors are not required to pay for the use of color figures.

Response:  All figures have been amended to have colours [Figure 5 -Line 349 ; Figure 6- Line 362; Figure 7- Line 395].

6. There may be a need for an exploratory analysis of the total information that the authors have so that the authors can create a general picture of the situation, such as a PCA or Pearson correlation of the major variables.

Response:  We all agree that the study is exploratory in nature, and that the data obtained is based on a small sample size. Future research can be suggested to extrapolate the database to find the correlation with the presence of significant microbes.

7. I think the discussion is too speculative. Analyze your own results to determine whether they are relevant or not.

Response:  Re-phrasing has be amended(Line 437-652)

Round 2

Reviewer 2 Report

Thank you for taking into account my previous comments. Currently, the manuscript reads well and the organization of the ideas appears more concise. I have no further comments to make.